# AnveshanaAI: A Multimodal Platform for Adaptive AI/ML Education Through Automated Question Generation and Interactive Assessment

## Abstract

We propose **AnveshanaAI**, an application-based learning platform for **artificial intelligence**. With AnveshanaAI, learners are presented with a personalized dashboard with streaks, levels, badges, and structured navigation across domains such as data science, machine learning, deep learning, transformers, generative AI, large language models, and multimodal AI, with scope to include more in the future. Through our portal, we design gamified tracking with points and achievements to enhance engagement and learning, while switching between **Playground, Challenges, Simulator, Dashboard, and Community** supports exploration and collaboration. Rather than using static question repositories like existing platforms, we ensure balanced learning progression through a dataset grounded in **Bloom's taxonomy**, with semantic similarity checks and explainable AI techniques improving transparency and reliability, along with adaptive, automated, and domain-aware assessment methods. The experiments depict broad dataset coverage, stable fine-tuning with reduced perplexity, and measurable gains in learner engagement. Together, these features illustrate how AnveshanaAI integrates adaptivity, gamification, interactivity, and explainability to support next-generation AI education.

The rapid growth of artificial intelligence (AI) and machine learning (ML) has created a strong demand for platforms that enable effective skill development and hands-on practice. Although existing coding environments such as CodeSignal (CodeSignal Team, 2025), StrataScratch (StrataScratch Team, 2025), and Exercism (Exercism Team, 2025) provide structured exercises in programming and data science, they fall short in addressing the unique requirements of AI/ML education. Unlike general coding tasks, AI/ML problem-solving requires not only algorithmic implementation but also conceptual reasoning, experimentation with models, and interpretation of results within dynamic contexts.

Prior research has explored areas such as automated question generation, adaptive assessment, and large-scale challenge design, but these efforts have largely remained fragmented. Current platforms lack an integrated ecosystem that unifies these components to effectively simulate real-world AI/ML tasks. In particular, three persistent gaps remain: the *limited support for automated generation of diverse and pedagogically meaningful challenges*, the *absence of robust mechanisms for fairness, adaptability, and scalability in practice-based learning*, and the *lack of simulation-driven and competitive features that can sustain learner motivation and long-term engagement*.

This study addresses these gaps by investigating the design of an integrated AI/ML practice platform that brings together automated question generation, adaptive assessment, and validation mechanisms. The work is guided by three **research questions (RQ)**:

- **RQ1:** How can automated question generation methods be adapted to produce high-quality, diverse, and skill-aligned challenges for AI/ML learners?

- **RQ2:** What mechanisms can be implemented to ensure fairness, adaptability, and scalability in AI/ML challenge-based learning platforms?

- **RQ3:** How can simulation and competitive features enhance the pedagogical effectiveness and long-term engagement of AI/ML learners within such platforms?

By framing the investigation around these questions, this study aims to lay the foundation for a next-generation practice-based AI/ML learning environment that balances scalability, quality, and learner engagement.

# 1 RELATED WORKS

## 1.1 PLATFORMS FOR LEARNING AND ASSESSMENT

Several platforms support practice-based learning and problem-solving in programming and data science. For example, StrataScratch (StrataScratch Team, 2025) provides analytical and algorithmic questions across SQL, data science, and software development, with filters for difficulty, companies, and industries. It also includes resources tailored to specific companies such as Accenture, Airbnb, Amazon, and Apple, and supports PostgreSQL, MySQL(Oracle Corporation, 1995), and Python-Pandas. Sigmoid Academy(Sigmoid Academy, 2024) hosts problem sets aimed at structured data-related practice, while Deep-ML(Deep-ML Team, 2024) curates collections of machine learning problems for benchmarking and skills evaluation.

While these platforms provide *curated problem sets* and structured practice opportunities, they lack mechanisms for adaptive question generation, difficulty calibration, and peer-driven feedback. This highlights the need for more personalized and scalable learning systems.

## 1.2 QUESTION GENERATION AND PEER ASSESSMENT SYSTEMS

Parallel research has investigated automated question generation and peer assessment platforms. Maarek and McGregor(Maarek & McGregor, 2020) proposed a peer feedback platform for programming artifacts that integrates software testing with peer assessment. The platform enables unit testing, scenario testing, and anonymity, enhancing both feedback quality and collaboration.

Recent work explores large language models for question generation. Doughty and Wan evaluated GPT-4(Radford et al., 2019) for generating multiple-choice questions (MCQs) (Doughty & Wan, 2023) in programming education, comparing 651 generated and 449 human-crafted MCQs Their findings suggest that LLMs can produce questions with clarity, alignment to Bloom's taxonomy (Bloom, 1956; Ghosh et al., 2024), and strong learning objective correspondence, thereby reducing educators' workload.

In difficulty estimation, Wang et al. introduced C-BERT(Zhang et al., 2020), a multimodal approach combining BERT(Reimers & Gurevych, 2019) and CodeBERT (Zhang et al., 2020) to jointly model problem text and code solutions. Experiments on Codeforces (Codeforces Team, 2024) and CodeChef (CodeChef Team, 2024)datasets demonstrated its superiority over baselines in estimating problem difficulty.

Domain-specific models have also been explored. proposed EduQG (Bulathwela et al., 2020), a model adapted from T5 (Maarek & McGregor, 2020) and fine-tuned on scientific text and educational question datasets. EduQG (Bulathwela et al., 2020) outperforms baseline approaches in generating pedagogically relevant and educationally sound questions, illustrating the benefit of domain-specific adaptation.

## 1.3 SUMMARY OF GAPS

In summary, existing platforms (e.g., StrataScratch (StrataScratch Team, 2025), Sigmoid Academy (Sigmoid Academy, 2024), Deep-ML(Deep-ML Team, 2024)) emphasize curated and static question repositories, while research on question generation highlights adaptive, automated, and domain-aware assessment methods. However, the integration of dynamic question generation, difficulty calibration, and community-driven peer feedback within practice platforms remains underexplored, presenting a promising research direction.

## 2 DATA CONSTRUCTION

The dataset underlying AnveshanaAI was designed to support scalable question generation and adaptive learning, ensuring both technical correctness and pedagogical rigor. Unlike conventional problem–answer corpora, the dataset integrates structured metadata that enables difficulty scaling, multimodal transformations, and curriculum-aware personalization.

### 2.1 SOURCES AND PREPROCESSING

We constructed a dataset of over **10,000 problem–answer pairs** across core domains of Artificial Intelligence and Machine Learning. Input sources included curated academic material (course notes, challenge repositories, and research references) as well as a seed set of human-authored tasks. All problems were standardized through preprocessing steps such as tokenization, chunking, and embedding, and were stored in a vectorized format for efficient retrieval. Each task was encapsulated as a *context package*, linked with metadata such as Bloom's taxonomy (Bloom, 1956; Ghosh et al., 2024)level and difficulty annotation.

### 2.2 SCHEMA AND AUGMENTATION

The dataset follows a structured schema with fields: **id**, **problem**, **answer**, **category**, **difficulty**, **tags**, and **bloom_level** (Bloom, 1956; Ghosh et al., 2024), ensuring semantic retrieval, traceability, and alignment with cognitive progression. To enhance coverage and diversity, two augmentation strategies were employed. First, **difficulty scaling** introduced rephrased problems, domain-shift variants, and edge cases, enriching the dataset with varying levels of challenge. Second, **cross-mode adaptation** transformed base problems into coding, simulation, debugging, and viva-style formats, thereby expanding task variety while preserving alignment with original concepts. Collectively, these strategies improved the dataset's robustness and pedagogical depth.

### 2.3 VALIDATION AND QUALITY ASSURANCE

Multiple validation layers were used to guarantee quality. Automated *LLM self-checks* filtered inconsistent problems, while *static validation* ensured syntactic correctness. For executable coding tasks, sandbox execution validated determinism and robustness. In parallel, rubric-based alignment ensured coverage across Bloom's taxonomy(Bloom, 1956; Ghosh et al., 2024) and balanced representation across difficulty levels.

### 2.4 DATASET CHARACTERISTICS

The final dataset comprises **10k+ entries**, spanning beginner to expert levels across categories such as Machine Learning, Deep Learning, Transformers, Generative AI, and Large Language Models. Each entry is enriched with metadata to support adaptive delivery, personalization, and multimodal task generation within the AnveshanaAI platform.

## 3 PLATFORM FUNCTIONALITIES

The proposed system caters to two types of users: (i) the **learners**, who solve challenges across multiple modes, and (ii) the **administrators/instructors**, who design, deploy, and monitor the challenges. Given these roles, we describe the major panels of the platform.

### 3.1 LEARNER PANEL

The learner panel provides an interactive and gamified experience through the following components:

1. **Landing Dashboard:** A personalized home page that greets learners with their current level, day streak, and accumulated points. It summarizes progress through metrics such as total challenges completed and learning paths explored. Gamification elements such as streaks, badges, and levels sustain engagement.

2. **Category Navigation:** Challenges are organized into structured categories including *Machine Learning*, *Deep Learning*, *Transformers*, *Generative AI*, *Large Language Models*, and *Multimodal AI*, enabling targeted exploration.

3. **Featured Challenges:** Highlighted tasks such as the *Neural Net Forward Pass* are showcased to promote trending or recommended challenges.

4. **Gamified Progress Tracking:** Learners can track their level, points, and streaks directly within the interface. This provides real-time reinforcement of continuous practice.

5. **Core Functionalities:** Quick access to the *Playground*, *Challenges*, *Simulator*, *Dashboard*, and *Community* through the top navigation bar ensures smooth mode switching.

## 3.2 ADMINISTRATOR PANEL

The administrator panel supports instructors and platform managers with three key functionalities:

1. **Challenge Design and Upload:** Admins can create new problems using a structured schema (problem, answer, difficulty, tags, Bloom level(Bloom, 1956; Ghosh et al., 2024)) or upload them via CSV/JSON. Augmentation pipelines (paraphrasing, difficulty scaling, cross-mode adaptation) enrich challenge diversity.

2. **Performance Analytics:** Dashboards summarize learner performance, highlighting difficult concepts, repeated errors, and engagement levels. Metrics such as accuracy, completion rate, and time-to-solution support adaptive feedback.

3. **Data Export and Integration:** Challenges, learner telemetry, and metadata can be exported for research or integrated into external LMS platforms.

## 3.3 SYSTEM ARCHITECTURE

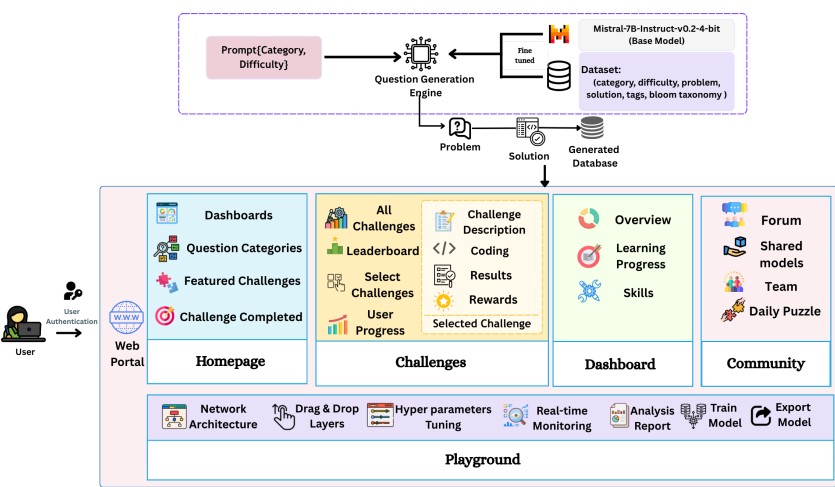

Figure 1: System architecture of the proposed platform.

The overall architecture (Figure 1) follows a modular design consisting of the Question Generation Pipeline (QGP), adaptive delivery engine, multimodal interaction layer, and analytics dashboard. Core technologies include fine-tuned LLMs for question generation, Docker(Docker Inc., 2013)[1]-based sandboxes for secure code execution, vector databases for context retrieval, and Whisper-based ASR for viva interaction. This modular approach ensures scalability, flexibility, and real-time interactivity.(Pyatkin et al., 2022)

---

[1]https://github.com/docker/docker-ce

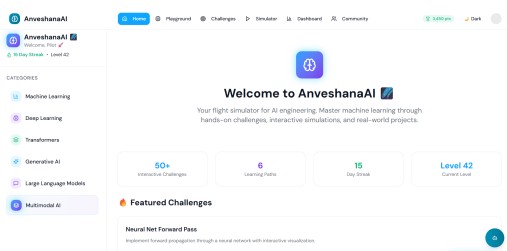

Figure 2: Landing Dashboard of the AnveshanaAI platform.

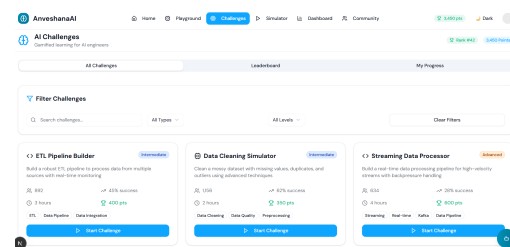

Figure 3: Challenges Interface showcasing interactive problem-solving modes.

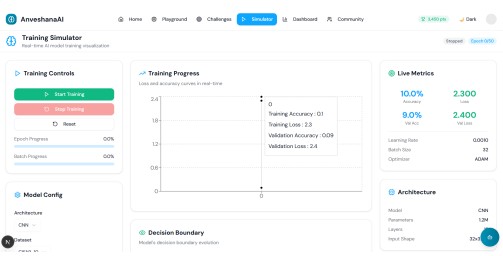

Figure 4: Simulation Lab for experimenting with AI/ML models and visualizations.

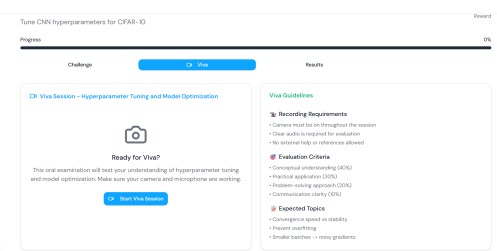

Figure 5: Viva Mode enabling oral-style Q&A using speech-to-text and LLM-driven questioning.

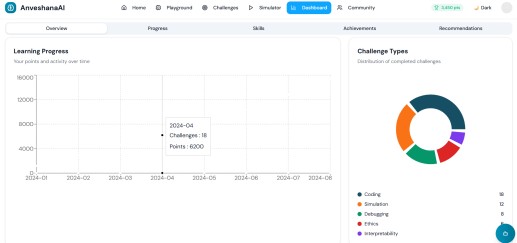

Figure 6: Dashboard showcasing personalized progress tracking, streaks, levels, badges.

## 4 METHODOLOGY

The development of **AnveshanaAI** follows a systematic methodology that integrates pedagogical design with robust technical implementation. The objective is to create an interactive and adaptive platform for learners, combining AI-driven assessment, coding challenges, and immersive simulations within a unified ecosystem.

At the core of the methodology lies the *system architecture*, which is designed as a modular, service-oriented. The frontend is built using **React with Vite** (Meta Platforms Inc., 2013)[1] for rapid rendering and responsiveness, while **Tailwind CSS** (Tailwind Labs, 2017)[2] ensures a consistent and visually appealing interface. The backend is powered by **Node.js and Express**(OpenJS Foundation, 2009)[3](ExpressJS Contributors, 2010)[4], which handle user management, session data, challenge execution, and analytics. A **MySQL database** (Oracle Corporation, 1995)[5] supports secure data storage, including user profiles, performance logs, and challenge metadata.

---

[2] https://github.com/tailwindlabs
[3] https://github.com/nodejs/node
[4] https://github.com/expressjs/express
[5] https://github.com/mysql/mysql-server

The *learning pipeline* begins with the user logging into the platform through the **Landing Dashboard**, which personalizes the experience by displaying recent activities, pending challenges, and progress metrics. Learners can then explore the **Challenges Interface**, which hosts problem sets across multiple domains and difficulty levels. Each challenge is connected to an automated evaluation engine that executes submitted code in a sandbox environment, ensuring fairness, security, and reproducibility.

To complement the problem-solving mode, the **Simulation Lab** provides interactive, scenario-driven exercises where learners can apply theoretical concepts in practical contexts. This includes system-level experiments, case-based simulations, and exploratory tasks that mimic real-world problem environments. The **Viva Mode** further extends the methodology by incorporating natural language interactions with an AI-powered evaluator, enabling assessment of conceptual clarity and reasoning in a semi-structured oral examination format. Finally, the methodology integrates **analytics and adaptivity** as core components. User performance is continuously tracked across challenges, simulations, and viva sessions. These data points are processed using machine learning techniques to generate personalized feedback, difficulty adjustments, and progress recommendations. This adaptive mechanism ensures that the platform not only evaluates learners but also supports their growth through data-driven guidance.

## 5 EXPERIMENTATION

The experimental phase was designed to validate three core aspects of the system: *(i)* the quality and representativeness of the constructed dataset, *(i)i* the performance of fine-tuned models in terms of optimization stability and predictive capability, and *(iii)* the interpretability of model outputs through explainability methods.

### 5.1 DATASET EVALUATION

The dataset was comprehensively analyzed along pedagogical, semantic, and annotation quality dimensions to ensure its reliability and utility for educational AI applications.

#### 5.1.1 MULTI-DIMENSIONAL ANNOTATION QUALITY ANALYSIS

We conducted a systematic evaluation of annotation consistency across three key dimensions: subject categories, difficulty levels, and Bloom's taxonomy (Bloom, 1956; Ghosh et al., 2024)classifications. Table 1 presents comprehensive quality metrics demonstrating the dataset's robust annotation framework.

Table 1: Multi-Dimensional Annotation Quality Metrics

| Annotation Dimension | Total Categories | Effective Categories | Entropy | Concentration Index | Sample Size |
|---|---|---|---|---|---|
| Category | 26 | 16.57 | 4.051 | 0.044 | 10,845 |
| Difficulty | 4 | 3.65 | 1.866 | 0.053 | 10,845 |
| Bloom Level | 6 | 5.84 | 2.546 | 0.011 | 10,845 |

The analysis reveals exceptional annotation quality across all dimensions. The category dimension demonstrates comprehensive topic coverage with 64% effective utilization (16.57/26 categories) and high entropy (4.051), indicating rich diversity without concentration bias (0.044). The difficulty dimension achieves near-complete utilization (91%) across all four levels, ensuring balanced representation from easy to expert-level questions. Most notably, the Bloom taxonomy(Bloom, 1956; Ghosh et al., 2024) dimension shows outstanding cognitive completeness with 97% effective utilization (5.84/6 levels) and minimal concentration (0.011), confirming comprehensive coverage of cognitive complexity levels.

#### 5.1.2 CROSS-DIMENSIONAL CORRELATION ANALYSIS

To assess the relationships between annotation dimensions, we computed Cramér's V (Cramér, 1946; Chen et al., 2025) correlation coefficients, which measure association strength between cat-

egorical variables. Figure 7 visualizes these relationships through a correlation heatmap, while Table 2 provides detailed quantitative analysis.

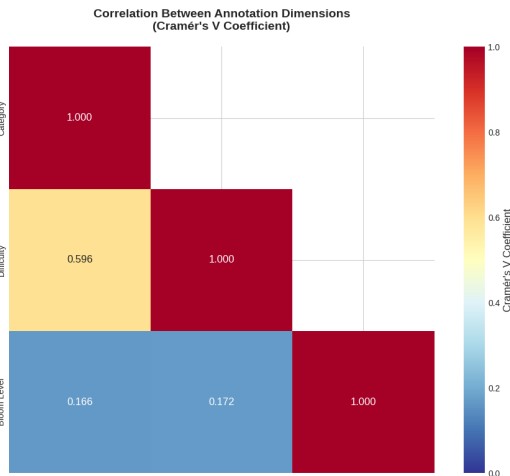

Figure 7: Cramér's V(Cramér, 1946; Chen et al., 2025) correlation heatmap showing associations between annotation dimensions. Values range from 0 (no association) to 1 (perfect association).

The strong category-difficulty correlation (0.596) validates systematic annotation patterns, demonstrating that certain subject domains naturally exhibit higher complexity. Conversely, the weak correlations between Bloom taxonomy and other dimensions (0.166-0.172) confirm that cognitive complexity operates independently of subject matter and perceived difficulty, aligning with established educational frameworks (Bloom, 1956).

Table 2: Cross-dimensional correlation analysis (Cramér's V coefficient)

| Dimension pair | Cramér's V | Strength | Educational implication |
|---|---|---|---|
| Category $\leftrightarrow$ Difficulty | 0.596 | Strong | Domain-specific complexity patterns support adaptive learning systems |
| Category $\leftrightarrow$ Bloom level | 0.166 | Weak | Independent cognitive assessment enables multi-dimensional evaluation |
| Difficulty $\leftrightarrow$ Bloom level | 0.172 | Weak | Cognitive complexity operates independently of perceived difficulty |

### 5.1.3 PEDAGOGICAL DISTRIBUTION ANALYSIS

Figure 8 illustrates the distribution of Bloom's taxonomy levels (Bloom, 1956; Ghosh et al., 2024) across four difficulty categories (*Easy, Medium, Hard, Expert*). The heatmap highlights that the dataset maintains a balanced representation of cognitive levels, with notable density in the mid-level categories of *Analyze*, *Apply*, and *Evaluate*. This ensures that learners are not restricted to rote memorization but are progressively challenged to apply, analyze, and reason through problems.

The heatmap in Figure 8 highlights that the dataset maintains balanced representation of cognitive levels, with notable density in mid-level categories of *Analyze*, *Apply*, and *Evaluate*. This ensures learners progress beyond rote memorization to higher-order thinking skills. The semantic similarity analysis (Figure 9) confirms that question-answer pairs cluster between 0.6-0.8 similarity, indicating strong contextual coherence without trivial repetition.

### 5.1.4 DATASET RELIABILITY ASSESSMENT

The evaluation highlights several key strengths of the proposed dataset. First, its **scale and coverage** are substantial, comprising 10,845 questions across 26 subject categories, thereby ensuring broad

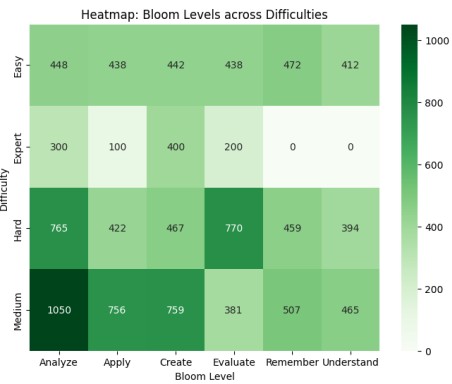

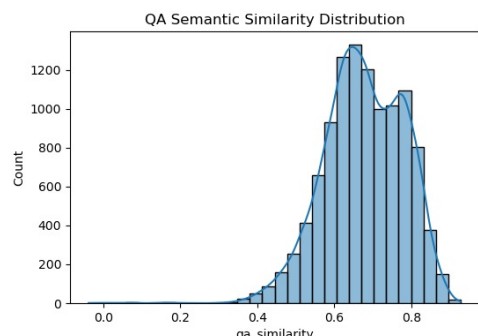

Figure 8: Heatmap showing the distribution of Bloom's taxonomy (Bloom, 1956; Ghosh et al., 2024) levels across difficulty categories.

Figure 9: Distribution of semantic similarity between question–answer pairs across the dataset.

applicability for machine learning–driven educational tasks. Second, the dataset demonstrates near-complete **cognitive completeness**, with 97% utilization of Bloom's taxonomy levels, which guarantees representation of the full spectrum of cognitive complexity. Third, the **balanced distribution** of annotations is evidenced by consistently low concentration indices ($\leq 0.053$), indicating the absence of bias toward particular categories or difficulty levels. Furthermore, the cross-dimensional correlations align with established theory, reinforcing the dataset's **pedagogical validity**. Finally, **semantic coherence** analysis shows that question–answer pairs achieve strong alignment, clustering between 0.6 and 0.8 similarity, while still maintaining sufficient diversity to avoid redundancy. Collectively, these findings confirm the dataset's **reliability** and its suitability as a foundation for adaptive, cognitively grounded AI learning platforms.

These metrics collectively establish the dataset's suitability for educational AI research, providing a robust foundation for developing and evaluating question-answering systems, difficulty prediction models, and adaptive learning algorithms.

## 5.2 FINE-TUNING PERFORMANCE

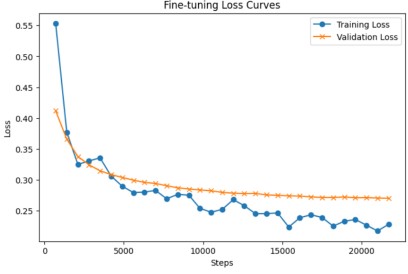

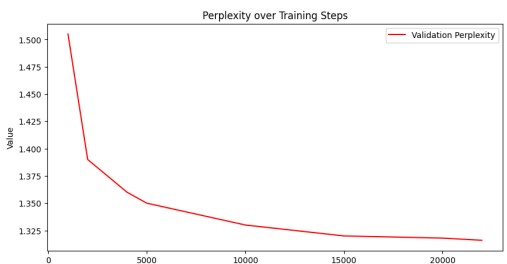

Figure 10: Training and validation loss curves observed during fine-tuning.

Figure 11: Validation perplexity over training steps.

The fine-tuned Mistral 7B model was evaluated using training–validation dynamics and perplexity. Figure 10 presents the training and validation loss curves. Both curves exhibit consistent downward trends, converging after approximately 15k steps, with no signs of severe overfitting. This demonstrates that the model effectively internalized the patterns in the dataset while maintaining strong generalization capabilities.

Validation perplexity(Liu et al., 2023) trends, shown in Figure 11, decreased steadily from around 1.5 to 1.3 over the course of training. This reduction demonstrates improved predictive capability and stable optimization, validating the effectiveness of the fine-tuning strategy.

| Token | Importance |
| --- | --- |
|  | 1.000 |
| explanation | 0.596 |
| research-quality | 0.474 |
| Generate | 0.465 |
| Bloom | 0.405 |
| problem | 0.352 |
| of | 0.348 |
| taxonomy | 0.331 |
| structure, | 0.314 |
| a | 0.313 |

Figure 12: XAI heatmap highlighting token importance.

## 5.3 Explainable AI Analysis

To probe the interpretability of the model, we performed a token importance analysis based on gradients on research-style prompts highlights the top ten most influential tokens, with darker shades indicating higher attribution scores.

The model consistently assigned high importance to semantically meaningful tokens such as *explanation*, *research-quality*, and *Bloom*(Bloom, 1956; Ghosh et al., 2024), confirming that its predictions are guided by contextually relevant information. This suggests that AnveshanaAI not only generates reliable outputs but also exhibits interpretable reasoning patterns aligned with educational objectives.

## 6 Results and Analysis

The results validate both **dataset quality** and **fine-tuning effectiveness** across dimensions of design, convergence, interpretability, and quantitative evaluation. The curated dataset of over **10,000 QA pairs** offers balanced coverage of **Bloom's taxonomy** (Bloom, 1956; Ghosh et al., 2024), strong **semantic diversity**, and reliable annotations. Fine-tuning **Mistral-7B** with **4-bit quantization** (Jiang & et al., 2023) yielded stable convergence and consistently reduced **perplexity**(Jelinek & Mercer, 1977; Liu et al., 2023), confirming efficient training without loss of performance.

Quantitative evaluation reported a low **perplexity**Liu et al., 2023 of **2.04**, demonstrating high fluency, and a **BERTScore F1**(Zhang et al., 2020) of **0.427** (**Precision**(Van Rijsbergen, 1979; Bronnec et al., 2024) = **0.289**, **Recall**(Van Rijsbergen, 1979; Bronnec et al., 2024) = **0.818**, indicating strong semantic coverage with extended explanatory richness. **Explainability analysis** further showed that the model attends to semantically relevant tokens, enhancing **interpretability** and **transparency** in reasoning.

Overall, these findings confirm that the integration of a **well-curated dataset** with **efficient fine-tuning** produces a model that is **fluent**, **interpretable**, and **pedagogically grounded**.

## 7 Conclusion

The combination of low **(Liu et al., 2023)**, high **recall(Van Rijsbergen, 1979; Bronnec et al., 2024)**, and **interpretable reasoning patterns** highlights both **methodological soundness** and **practical applicability**. While **precision(Van Rijsbergen, 1979; Bronnec et al., 2024)** remains lower due to elaborative outputs, this is beneficial in **educational contexts** where detailed explanations aid learner understanding. The dataset's **balanced coverage** across taxonomy levels ensures robust evaluation of higher-order reasoning, and the use of **quantization** establishes computational efficiency. Collectively, these results demonstrate that **AnveshanaAI** serves as both a **reliable dataset** and an **effective platform** for **transparent**, **adaptive**, and **educationally aligned AI systems**.

## Reproducibility Statement

We provide details necessary to reproduce our results as follows:

- **Experimentation:** Described in Section 5 of the main text.
- **Data generation:** We constructed a dataset based on category and difficulty of ~10,000 Q/A pairs later refined through filtering and semantic similarity scoring. The dataset is publicly available at `https://huggingface.co/datasets/t-Shr/Anveshana_AI/blob/main/data.csv`.
- **Model training:** We fine-tuned the Mistral-7B v0.1 model (4-bit quantization, LoRA adapters) on our generated dataset using HuggingFace's Trainer. Training was performed with batch size 2 per device, learning rate $2 \times 10^{-5}$, weight decay 0.01, and 5 epochs. Checkpoints were saved every 500 steps (max 3 retained), while evaluation was run every 700 steps with logging enabled. Mixed precision (FP16) training was used on a single 20GB GPU.

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
