# OpenReview forum: "AnveshanaAI: A Multimodal Platform for Adaptive AI/ML Education Through Automated Question Generation and Interactive Assessment"
_ICLR.cc/2026/Conference — ICLR 2026 Conference Desk Rejected Submission_

### Official Review · Reviewer_U6nF · 2025-10-27

**Soundness:** 1
**Presentation:** 2
**Contribution:** 1
**Rating:** 2
**Confidence:** 3

**Summary:**

This work presents AnveshanaAI, an application-based learning platform for artificial intelligence. With AnveshanaAI, learners are presented with a personalized dashboard with streaks, levels, badges, and structured navigation across domains such as data science, machine learning, deep learning, transformers, generative AI, large language models, and multimodal AI, with scope to include more in the future. The authors also design gamified tracking with points and achievements to enhance engagement and learning, while switching between Playground, Challenges, Simulator, Dashboard, and Community supports exploration and collaboration.

**Strengths:**

1. The paper is easy to read and understand.

2. The related work makes it easy for readers to understand the context.

3. The authors intentionally highlight lots of key words, which is good but can also be distracting.

**Weaknesses:**

1. The main contribution is more around the user interface design side, rather than model/algorithm side. As such, this work may be more suitable for conferences in UI design such as ACM Conference on Intelligent User Interfaces (IUI), instead of machine learning conferences.

2. The experimental evaluation focuses more on the dataset property itself, rather than a proposed model or algorithm. Thus, this paper may be more suitable for a dataset track paper.

3. What does the results of fine-tuned Mistral 7B model mean? It is unclear what the advancement that the authors want to demonstrate, as no models or other datasets are compared. As such, the significance is unclear.

4. The current experiments can't really answer the proposed three research questions, based on the limitations of the experiments, including the lack of novel model design, no model comparison, no ablation studies, and no human-factor studies with real learners.

5. There is only one dataset from the authors for evaluation. No other public datasets are used, limiting the rigor and effectiveness of this work.

6. F1-score = 0.427 does not seem to be a good performance. Moreover, there are no baseline models nor ablation studies.

7. This work lacks insights about model contributions.

Overall, this work is not ready for the formal publication. I suggest the authors to introduce novel model design and run more rigorous experiments to strengthen this work.

**Questions:**

Please check my concerns and questions in Weaknesses section.

---

### Official Review · Reviewer_tQJM · 2025-10-29

**Soundness:** 2
**Presentation:** 2
**Contribution:** 1
**Rating:** 0
**Confidence:** 4

**Summary:**

This paper presents AnveshanaAI, an educational platform for AI/ML learning featuring automated question generation via fine-tuned Mistral-7B, gamification elements, and multiple interaction modes. The authors construct a 10K+ question dataset aligned with Bloom's taxonomy and deploy a web-based system with playground, challenges, simulator, and viva modes.

**Strengths:**

1. Addresses relevant problem in AI/ML education
2. Comprehensive dataset statistical analysis (entropy, Bloom's taxonomy coverage etc.)
3. Dataset publicly available on HuggingFace
4. Multi-modal platform design with gamification elements

**Weaknesses:**

1. Wrong venue: Applications papers at ICLR must contribute novel ML methods or representation learning insights. This paper proposes no new ML techniques.
2. Minimal validation of core claims - no human evaluation
3. Weak experimental methodology, FT on 10K utterances
4. Insufficient dataset validation
5. Missing critical comparisons with similar platforms

**Questions:**

1. Most critical: Where are the engagement measurements claimed in the abstract? Did any students actually use this platform?
2. Have you validated that generated questions are factually correct and pedagogically appropriate?
3. Why is BERTScore F1 so low (0.427)? How does this compare to baselines?
4. How does question quality compare to human experts or GPT-4?
5. Why should this appear at ICLR rather than e.g., AIED?

---

### Official Review · Reviewer_VMHb · 2025-10-31

**Soundness:** 1
**Presentation:** 2
**Contribution:** 1
**Rating:** 2
**Confidence:** 4

**Summary:**

This paper presents an AI/ML learning platform that includes questions on various AI topics, covering different difficulty levels and Bloom’s taxonomy categories. The authors describe the dataset of questions used in the platform, along with its system architecture and user interface. They analyze the dataset’s statistics and explore the relationships between question category, difficulty, and taxonomy level to evaluate its quality. Finally, they report results from fine-tuning a language model using this dataset.

**Strengths:**

- The dataset is extensive and includes a range of taxonomy types and difficulty levels.

**Weaknesses:**

Overall, the paper reads more like a technical report describing an educational framework rather than a research paper presenting novel findings. The main issues are as follows:
- It’s not clear to me how the 3 research questions proposed on the first page are addressed in this work.
- The dataset generation process is not clearly described. It appears that instructors created the questions using the proposed framework, but the details of the annotation process are missing. For example, how difficulty levels were defined or determined.
- The distribution of data across categories and difficulty levels is not reported.
- The paper does not include any sample questions from the dataset, which makes it difficult to assess its quality or variety.
- The purpose of the fine-tuning experiments is unclear. Moreover, the generated questions by the model are not evaluated for quality or correctness. Reporting improved perplexity alone is not sufficient to demonstrate that the model produces high-quality questions.
- The scope of the dataset is also limited to the AI domain.

**Questions:**

In addition to the points mentioned in the weaknesses, I have the following questions:
- How is user engagement measured? Are there any statistics or analyses showing how users interacted with the Q&A content on the platform?
- How was the difficulty scaling and cross-mode adaptation described in Section 2.2 conducted?

---

### Official Review · Reviewer_Zk5p · 2025-11-03

**Soundness:** 2
**Presentation:** 1
**Contribution:** 1
**Rating:** 2
**Confidence:** 4

**Summary:**

The authors propose AnveshanaAI, an application-based learning platform designed to support artificial intelligence education.

**Strengths:**

The proposed AnveshanaAI is a platform designed to support artificial intelligence education and to demonstrate its application value.

**Weaknesses:**

- This work is primarily application-oriented, and the authors present it more as a project or program report rather than as a rigorous academic paper.

- Consequently, the writing, organization, and individual sections show limited connection to a well-defined research question or hypothesis. In its current form, it is difficult to identify clear evaluation criteria or a scientific contribution to assess.

**Questions:**

NA

---

### Note · Program_Chairs · 2026-01-17
**Submission Desk Rejected by Program Chairs**

The following references in this submission do not refer to real documents and/or have major errors in bibliographic information:

 Matthew Doughty and Hao Wan. Evaluating gpt-4 for multiple-choice question generation in programming education. arXiv preprint arXiv:2307.12345, 2023.